# Perfluorooctanoic Acid (PFOA) Exposure Compromises Fertility by Affecting Ovarian and Oocyte Development

**DOI:** 10.3390/ijms25010136

**Published:** 2023-12-21

**Authors:** Han Zhang, Lulu Han, Lijun Qiu, Bo Zhao, Yang Gao, Zhangjie Chu, Xiaoxin Dai

**Affiliations:** School of Fisheries, Zhejiang Ocean University, Zhoushan 316022, China; zhanghan@zjou.edu.cn (H.Z.); hanlulu@zjou.edu.cn (L.H.); qiulijun0412@126.com (L.Q.); zhaobo1050@163.com (B.Z.); gaoyang-82@163.com (Y.G.); czj0501@zjou.edu.cn (Z.C.)

**Keywords:** zebrafish, fertilization capability, oocytes, perfluorooctanoic acid

## Abstract

PFOA, a newly emerging persistent organic pollutant, is widely present in various environmental media. Previous reports have proved that PFOA exposure can accumulate in the ovary and lead to reproductive toxicity in pregnant mice. However, the potential mechanism of PFOA exposure on fertility remains unclear. In this study, we explore how PFOA compromises fertility in the zebrafish. The data show that PFOA (100 mg/L for 15 days) exposure significantly impaired fertilization and hatching capability. Based on tissue sections, we found that PFOA exposure led to ovarian damage and a decrease in the percentage of mature oocytes. Moreover, through in vitro incubation, we determined that PFOA inhibits oocyte development. We also sequenced the transcriptome of the ovary of female zebrafish and a total of 284 overlapping DEGs were obtained. Functional enrichment analysis showed that 284 overlapping DEGs function mainly in complement and coagulation cascades signaling pathways. In addition, we identified genes that may be associated with immunity, such as LOC108191474 and ZGC:173837. We found that exposure to PFOA can cause an inflammatory response that can lead to ovarian damage and delayed oocyte development.

## 1. Introduction

Since the 1950s, Per-and polyfluoroalkyl substances (PFASs) have been widely used. They have been used in industrial and consumer products such as paint, makeup, textiles, carpeting, and other daily use items [1]. The two most representative and widely used PFASs are perfluorooctanesulfonic acid (PFOS) and perfluorooctanoic acid (PFOA).

PFOA is a long-chain perfluorinated chemical that has been shown to have non-negligible characteristics of non-degradability, leading to its difficulty in being eliminated efficiently by conventional wastewater treatment plants (WWTPs) [2]. Although PFOA has already been classified as a newly emerging persistent organic pollutant and many countries across the world have voluntarily ceased production [3,4], it can still be detected in various environmental media, including air, water, food, plants, animals and even in humans [5]. Most PFOA that cannot be effectively removed is released into aquatic environments and distributed in the matrix of natural water, aquatic animals, and plants [6]. Moreover, PFOA can be transferred and biomagnified through the food chain and presents a risk for high-trophic-level consumers [7]. Among seven Chinese river basins, the maximum average concentration of PFOA was in the Yangtze River Basin (58 ng/L), while the lowest concentration was in the Songhua River Basin (<1 ng/L) [8]. Another study investigated the level of PFOA contamination in twenty-five samples of fresh fillet of five widely consumed fish species purchased from large retailers in Italy, finding that PFOA was present in all samples, with concentrations up to 487 ng/kg (mean = 75 ng/kg) [9].

PFOA can be rapidly absorbed but is excreted by organisms with difficultly. In humans, the half-life of PFOA has been estimated at 3.8 years [10,11,12]. Previous reports have proved that PFOA exposure can lead to reduced fetal weight, reduced postnatal survival, delays in postnatal growth and development in offspring in pregnant mice [13]. In addition, PFOA has been shown to cause toxic effects on the female reproductive system by entering the follicular fluid through the blood–egg barrier [14]. In recent years, PFASs have received increasing public attention due to their adverse effects on humans and wildlife.

The ovary is a functional organ of the female reproductive system which releases a mature oocyte for fertilization [15]. During oogenesis in zebrafish, the oogenic cells come to a standstill at the diplotene stage of prophase during meiosis I. At this time, the oocyte is known as the germinal vesicle (GV stage) [16]. The oocyte is then stimulated by gonadotropin to resume meiosis I and germinal vesicle breakdown (GVBD) [17]. The oocyte re-arrests at the metaphase of meiosis II to await fertilization. Studies have demonstrated that pregnant mice exposed to PFOA showed significantly disrupted follicle development and inhibited oocyte maturation [14]. PFOA exposure also inhibits the secretion of ovarian hormones, which may disrupt ovarian function. A previous study demonstrated radiolabeled perfluorooctanoic acid (14C-PFOA) can accumulate in oocytes, suggesting PFOA may have adverse effects on early embryonic development and offspring health [18].

During ovulation, the ovary experiences damage and associated wound healing, exposing the ovaries and fallopian tubes to high levels of hormonal and inflammatory agents and causing an inflammatory-like reactions [19,20]. Mucins are high molecular weight biomolecules generated by the secretory epithelial cells that line the ducts and lumens of the human body. The key role of mucins is to keep epithelial surfaces hydrated, which is necessary for the lubrication and effective operation of ducts and passageways [21]. Mucins are also involved in the protection of the epithelial cells from infections and injuries [22]. Moreover, large amounts of inflammatory cytokines have been demonstrated to upregulate mucin expression [23].

In the present study, we investigated the adverse impact of PFOA on the fertility of zebrafish. We evaluated the oocyte quality by investigating ovary morphology, germinal vesicle breakdown, and fertilization ability. We also investigated the transcriptome of zebrafish ovaries to verify the potential mechanism of PFOA exposure on fertility. We observed that PFOA exposure caused ovarian damage and through an inflammatory response, delayed oocyte development.

## 2. Results

### 2.1. Acute Toxicity Test of PFOA on Zebrafish

According to Koch’s method, the LC50 and 95% confidence intervals of adult zebrafish exposed to 0, 250, 300, 350, 400, 450, and 500 mg/L of PFOA solution for 24 h, 48 h, 72 h, and 96 h were determined and are shown in Table 1.

### 2.2. PFOA Exposure Compromises the Fertility

To examine if PFOA would impair fertility in zebrafish, we compared fertilization rates and hatching rates of female zebrafish administered with different doses of PFOA (Table 2). Most control eggs could be fertilized and hatch to the protruding mouth stage, while the fertilization rates and hatching rates of PFOA-exposed zebrafish decreased in a dose-dependent manner (Figure 1A,B). The fish exposed to a dose of 100 mg/L PFOA exhibited a significantly lower fertilization rate (51.67 ± 1.20%) and hatching rate (56.74 ± 1.43%) compared with controls (78.00 ± 2.65%; 88.67 ± 2.19%) (Figure 1A,B). Thus, this dose (100 mg/L) was chosen for subsequent investigations.

In order to investigate the effects of PFOA on early embryonic development, we analyzed the deformation rates of the embryos. Our findings suggest that PFOA exposure leads to pericardial edema and spinal curvature in embryos (Figure 1C). The frequency of abnormally developed embryos in the group exposed to PFOA (13.61 ± 1.41%) was significantly higher than that in the control group (0.33 ± 0.33%) (Figure 1D).

### 2.3. Effect of PFOA on Ovary Morphology and Oocytes Maturation

To determine the possible reason for the failure of fertilization, we calculated GSI, performed ovarian sections, and counted oocytes at different times. We found the GSI of female zebrafish exposed to PFOA for 15 d (3.68 ± 1.07%) was significantly lower than that in controls (15.78 ± 2.32%) (Figure 2C).

In the ovaries of the control group, oocytes were connected by the gonadosomatic tissue and edged with follicular cells. However, the vacuolation of gonadosomatic tissue and a loss of contact between the oocytes and follicular cells were observed in zebrafish ovaries exposed to PFOA (Figure 2A). Moreover, the anomaly rates increased in a dose-dependent manner (Figure 2B).

After exposing female zebrafish to PFOA for 15 days, the number of primary growth stage oocytes increased significantly (19.75 ± 1.97%) compared with the control group (13.30 ± 0.17%) (Figure 2D). Furthermore, the study also revealed the number of oocytes in the pre-vitellogenic, vitellogenic, and maturation phases (Figure 2E–G) (The specific values are shown in Table 3). Interestingly, the proportion of mature eggs was found to be significantly lower (16.24 ± 1.55%) in the treatment group than in the control group (26.27 ± 1.31%) (Figure 2H). In conclusion, this study found that exposure to PFOA had a negative impact on zebrafish ovaries, resulting in a reduced number of mature oocytes.

To further examine the potential negative effects of PFOA-induced ovarian damage on oocyte maturation, an in vitro culture of oocytes was performed. Our findings suggest that although PFOA exposure may slightly inhibit oocyte maturation in the short term, the difference was not significant. However, after 15 days of exposure, the maturation of oocytes (39.45 ± 4.00%) was significantly lower than that of the control group (53.89 ± 1.11%) (Figure 2I). In addition, we examined the survival rate after in vitro culture and observed that the survival rate was high in the control group (73.89 ± 2.00%) and the treatment group (77.78 ± 1.47%) (Figure 2J).

### 2.4. Transcriptome Analysis Identified the Potential Mechanism of PFOA-Caused Abnormal Fertility

We conducted a transcriptome analysis on zebrafish to study the potential mechanism of abnormal fertility caused by PFOA. The raw data were obtained using the Illumina NovaSeq6000 (Illumina, San Diego, CA, USA) high-throughput sequencing platform. To ensure high-quality and clean data, we removed the linker sequence and poor-quality reads from the raw data. Table 4 shows the output and quality of the high-quality, clean data obtained after filtering. The transcriptome sequencing results are excellent, which ensures the accuracy of the subsequent study.

To identify the DEGs between the ‘Control’ and ‘Treatment’ groups, the data were filtered using specific criteria: |log2(Fold Change)| > 1 and q value < 0.05. The results reveal that 284 genes were differentially expressed between the two groups. Out of these, 180 genes were upregulated, while 104 were downregulated. To visually represent the significant DEGs, a volcano plot was created. In this plot, the significant DEGs are indicated by red and blue dots, while non-differentially expressed genes are represented by black dots (Figure 3A).

To further classify the expression patterns of DEGs, we obtained the clustering results of DEGs (Figure 3B) by performing a hierarchical cluster analysis on the chosen DEGs and grouping genes with identical or comparable expression behavior. These DEGs can also be classified into three ontologies: biological process (7174 terms), molecular function (973 terms), and cellular component (814 terms). The top 40 enriched GO terms (up and down) in three categories are presented in Figure 4. Finally, we also performed an enrichment analysis of the KEGG pathways, and the top 40 KEGG enrichment pathways are shown in Figure 5. The most enriched pathway was “complement and coagulation cascades” (14 DEGs).

According to the transcriptome data, two immune-related genes were selected. The expression levels of immune-related genes in the two groups were compared and assessed. The results of FPKM indicate that two genes were significantly expressed (Figure 6). The expression levels of LOC108191474 and ZGC:173837 were significantly higher in PFOA-exposed groups than in control groups.

## 3. Discussion

PFOA is a persistent and bioaccumulative environmental pollutant which has come under increasing scrutiny due to its hazardous effects on human health, particularly the reproductive system. A prior investigation revealed that PFOA exposure hampered follicle development and ovarian function in mice [14]. However, whether zebrafish exposure to PFOA has negative effects on fertility has not been defined. Therefore, we conducted experiments to investigate how PFOA exposure affects the fertility of female zebrafish. The results demonstrate that PFOA exposure to female zebrafish had negative effects on their reproductive parameters, histological structures, and oocyte maturation rate.

According to earlier research, the PFOA 96-h LC50 values in zebrafish were around 500 mg/L [24,25,26]. In another study, the PFOA 96-h LC50 for larval rainbow trout was 730 mg/L [27]. Our acute toxicity assay improved on Ahmed’s 2013 methodology and found that the PFOA 96-h LC50 in zebrafish (4–5 months of age) was about 375.578 mg/L [28].

Our experiment aimed to investigate the impact of PFOA exposure on the fertility of female zebrafish. To assess this, we used fertilization rate, deformity rate, and hatching rate as biomarkers. Our findings reveal that PFOA had adverse effects on the fertility of female zebrafish. This was demonstrated by a decrease in fertilization and hatching rates, as well as an increase in malformation rates. Previous studies have also reported similar negative impacts of PFOA on the reproductive capacity of zebrafish, including reduced hatching rates and egg production [29].

To determine how PFOA exposure affects the fertility of female zebrafish, GSI was utilized as a crucial biomarker to assess the toxic effects on whole gonads. We observed that PFOA exposure significantly reduced GSI, which suggests that ovarian lesions such as vacuolation and gonadal–somatic tissue lysis. In a previous report, PFOA exposure was reported to alter ovary weight and possibly reduce female fecundity [30]. In addition, we also observed a decrease in the number of mature oocytes. We speculate that this may have contributed to the reduction in the hatching rate. Previous studies have reported that PFOA can cross the blood–follicle barrier and is more likely to come in contact with and damage oocytes directly, which suggests PFOA is linked to poor fertility [31,32]. In addition, animal experiments have indicated that PFOA exposure causes a decline in the number of follicles by impairing the meiosis of oocytes [33].

To further verify whether PFOA exposure impedes oocyte maturation, an in vitro culture of oocytes was performed. We observed that PFOA exposure significantly reduced the maturation rate of oocytes. Previous studies have shown that PFOA hinders the maturation rate of mouse oocytes by triggering mitochondrial and DNA damage [34]. Furthermore, when PFOA was present throughout pregnancy, it greatly reduced oocyte maturation capability [14]. These findings are consistent with our experimental results. We hypothesize that the decrease in the number of mature oocytes in zebrafish leads to the decrease in their fertilization rate.

Finally, to verify the potential mechanism of PFOA exposure on fertility, we investigated the transcriptome of zebrafish ovaries. Our analysis of DEGs in the transcriptome data revealed that these DEGs were primarily enriched in the complement and coagulation cascades pathway. Additionally, we found that two important immune-related genes (LOC108191474, ZGC:173837) were upregulated in the group exposed to PFOA.

The complement system, which protects the host against the invasion and proliferation of various pathogens, is an integral part of the innate immune system and the first line of defense against microbial invaders [35]. In this study, the pathway in which the DEGs were most significantly enriched was the complement and coagulation cascade pathway. Triggered by an initial condition such as bacterial infection, immune cells can release a wide range of inflammatory mediators, including cytokines, chemokines, and complement activation products [36]. The persistent inflammation can eventually lead to tissue/organ damage and death. The transcriptomic data suggest that fourteen genes in the complement and coagulation cascade pathway are upregulated in expression. These results suggest that ovarian damage in zebrafish may be due to an inflammatory response.

Mucins regulate cell adhesion and signaling, the epithelium’s renewal and differentiation, and immune suppression [37]. In the present study, we found that the expression of LOC108191474 (mucin-17-like isoform X1) and ZGC:173837 (mucin-19 isoform X2) were significantly higher in the PFOA-treated ovaries. We speculate that the elevated expression of these two genes is associated with ovarian pathology. In addition, one study reported that MUC17 induces cell arrest through activation of the p38 pathway [38]. Therefore, we speculate that the increase in LOC108191474 expression may be related to the decreased oocyte maturation rate and reduced number of oocytes in our experiments.

## 4. Materials and Methods

### 4.1. Animals

Wild-type zebrafish (*Danio rerio*) of the AB strain were used in this study. They were maintained with dechlorinated tap water at 26 ± 1 °C, in a 14 h/10 h light-dark cycle. The fish were fed twice a day and were acclimatized for 30 days before the experiments.

### 4.2. Exposure Regimens

Female zebrafish were randomly assigned and exposed to nominal concentrations of 0, 250, 300, 350, 400, 450, and 500 mg/L PFOA (purity 96%, Sigma-Aldrich, St. Louis, MO, USA) for 96 h to determine the LC50 [39]. The solvent of PFOA was water. Zebrafish behavior and mortality were monitored and recorded every 8 h.

The highest exposure group concentration was set at 1/4 of the 96 h LC50 to ensure that zebrafish did not die during the experiment. Then, a stock solution of PFOA (1 g/L) was prepared and diluted to experimental concentrations (0, 25, 50, 75 and 100 mg/L). Thirty females were randomly assigned to five groups and exposed for 15 days. Every 48 h, half the water was renewed with water containing the specified nominal PFOA concentrations. After exposure, we examined the hatching rate, fertilization rate, and deformity rate of the zebrafish. We found significant changes in the hatching rate, fertilization rate, and deformity rated at a concentration of 100 mg/L. Therefore, this dose (100 mg/L) was chosen for subsequent investigations.

Finally, female zebrafish were randomly divided into two groups: control (*n* = 120) and treatment (*n* = 120, 100 mg/L). Three biological replicates were used for each of the two groups. The exposure lasted 15 days.

### 4.3. Sample Collection

During the experiment, we collected samples from the fish on the fifth, tenth, and fifteenth days of exposure. On each occasion, we randomly selected two fish from each group (*n* = 12) and euthanized them. We immediately removed the ovaries from the fish and weighed them to calculate the gonad-somatic index (GSI = gonad weight × 100/body weight). Following this, the ovaries were fixed in Bonn’s solution at 4 °C for 24 h to examine any histopathological changes. We then randomly selected three fish from each group (*n* = 18) and placed their ovaries in −80 °C for RNA extraction. Additionally, we removed some ovaries (*n* = 18) for in vitro culture.

### 4.4. Reproductive Indices

After exposure, three females from each group and six untreated males were placed in a single mating tank (18 × 8.5 × 15 cm) to mate. Then, three-hundred healthy fertilized eggs were randomly selected from each group and incubated in an incubator at 28 °C for 72 h. After 24 h, dead embryos were removed and the fertilization rate was calculated. The embryos continued to incubate for two days to evaluate the hatching and deformity rates.

### 4.5. Histological Analysis

The oocyte development of zebrafish has been divided into five stages: I (primary growth), II (cortical alveolus), III (vitellogenin), IV (maturation), and V (mature egg). Sections in each group were observed under the microscope (100× and 400×). The percentage of oocytes at each stage was determined by dividing the number of oocytes at each stage by the total number of oocytes observed in 5 randomly selected fields of view at 100× magnification using light microscopy.

### 4.6. In Vitro Culture of Zebrafish Oocytes

The ovary was quickly removed and placed into a special oocyte medium (OCM). The OCM consisted of 90% Leibovitz’s L-15 medium (Solarbio, Beijing, China), 0.5 mg/mL bovine serum albumin (BSA, Amresco, Solon, OH, USA) and 3% Penicillin-Streptomycin (PS, Gibco, Waltham, MA, USA). Intact follicles about 0.625 mm in diameter were isolated from ovarian fragments and then placed into the oocyte maturation medium (OMM, 90% Leibovitz’s L-15 medium, 0.5 mg/mL BSA, 1% PS and 1 ug/mL 17α-20β-dihydroxy-4 pregnen-3-one) (17α-20β-dihydroxy-4 pregnen-3-one, DHP, Medbio, Shanghai, China). After incubation at 26 °C for 24 h, the oocytes’ maturation and survival rates were examined. To determine the oocytes’ maturation, they were stained away from light with 20 umol/L brilliant cresyl blue (Yuanye, Shanghai, China) for 150 min, after which, mature cells were stained blue. Moreover, to calculate the survival rate, the oocytes were stained with 0.4% Trypan blue (Sigma-Aldrich, St. Louis, MO, USA) at room temperature for 5 min, and any dead cells were stained blue.

### 4.7. Transcriptome Sequencing

Total RNA from the ovary samples was extracted using the RNA extraction kit (TaKaRa, Code: NO. 9767, Beijing, China) following the manufacturer’s instructions. The quality and concentration of RNA were evaluated using 1% agarose gel electrophoresis and Nanodrop (Thermo Fisher, Carlsbad, CA, USA). Subsequently, the integrity of the RNA was evaluated using the RNA Nano 6000 Assay Kit of the Bioanalyzer 2100 system (Agilent Technologies, Santa Clara, CA, USA). A total of 1 µg RNA per sample was used for cDNA synthesis and RNA-seq. Sequencing libraries were generated using NEBNext^®^ UltraTM RNA Library Prep Kit for Illumina^®^ (NEB, San Diego, CA, USA) following manufacturer’s recommendations and index codes were added to attribute sequences to each sample. In order to select cDNA fragments of 250~300 bp in length, the library fragments were purified with AMPure XP system (Beckman Coulter, Beverly, MA, USA). The constructed DNA template was enriched by PCR amplification (15 cycles). Finally, the PCR products were purified (AMPure XP system) and the library quality was assessed on the Agilent Bioanalyzer 2100 system. The clustering of the index-coded samples was performed on a cBot Cluster Generation System using TruSeq PE Cluster Kit v3-cBot-HS (Illumia) according to the manufacturer’s instructions. After cluster generation, the library preparations were sequenced on an Illumina Novaseq platform and 150 bp paired-end reads were generated.

### 4.8. Differentially Expressed Genes and Enrichment Analysis

To obtain clean reads, the raw-paired end reads from all transcriptomes were cleaned by removing adapter contamination, low-quality sequences (reads with over 10% unknown base pairs ‘N’), and empty reads. The differentially expressed genes (DEGs) were calculated according to the negative binomial distribution test performed using the DESeq2 R package. To standardize the expression variations between the various samples, the threshold value was set to |log2(Fold Change)| > 1 and q value < 0.05. According to these criteria, we searched for differently expressed genes (DEGs) between the two groups. DEGs were compared with the Gene Ontology (GO) and the Kyoto Encyclopedia of Genes and Genomes (KEGG) databases.

### 4.9. Statistical Analysis

All data were analyzed using SPSS 23 for significant differences. The level of statistical significance (*p* < 0.05) was assessed by one-way ANOVA. Where appropriate, data are given as the mean ± SEM.

## 5. Conclusions

Our study examined the effects of a newly emerging persistent organic pollutant (PFOA) on the fertility of zebrafish. We found that zebrafish exposed to 100 mg/L of PFOA for 15 days experienced impaired fertility, as evidenced by reduced fertilization and hatching rates. Additionally, we observed that PFOA exposure increased the rate of offspring deformities. We sequenced the transcriptome of the ovary of female zebrafish, and a total of 284 overlapping DEGs were obtained. Functional enrichment analysis showed that the pathway that exhibits the greatest degree of enrichment is the complement and coagulation cascades signaling pathway. In addition, we identified genes that may be associated with immunity, such as LOC108191474 and ZGC:173837. We found these reproductive changes could be attributed to ovarian damage and a decrease in the percentage of mature oocytes. PFOA is rapidly absorbed by organisms but is difficult to excrete, and its presence in aquatic ecosystems can have a cascading effect on the food chain, potentially leading to human exposure. Therefore, it is important to study the toxicity of PFOA in fish. Previous studies have shown that PFOA has reproductive toxicity. However, the potential mechanism of PFOA exposure on fertility remains unclear. We found exposure to PFOA can cause an inflammatory response that can lead to ovarian damage and delayed oocyte development. This study’s findings provide valuable insights that can contribute to future research on the potential mechanisms of PFOA exposure on fertility.

## Figures and Tables

**Figure 1 ijms-25-00136-f001:**
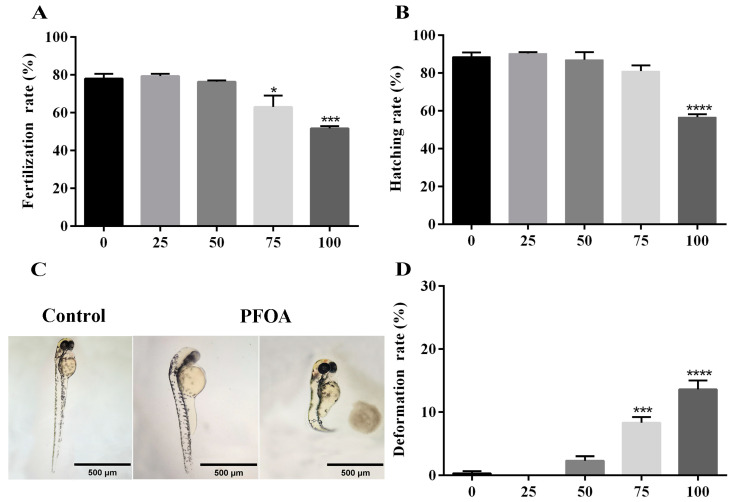
Effects of PFOA on fertility. (**A**) Fertilization rates at different concentrations. (**B**) Hatching rates at different concentrations. (**C**) Representative images of deformed eggs showing normal, pericardial edema and spinal curvature, respectively. (**D**) Deformation rates at different concentrations. Data are presented as mean percentage (mean ± S.E.M.) of at least three independent experiments (* *p* < 0.05, *** *p* < 0.001, **** *p* < 0.0001); the vertical bar represents the standard error of mean (SEM).

**Figure 2 ijms-25-00136-f002:**
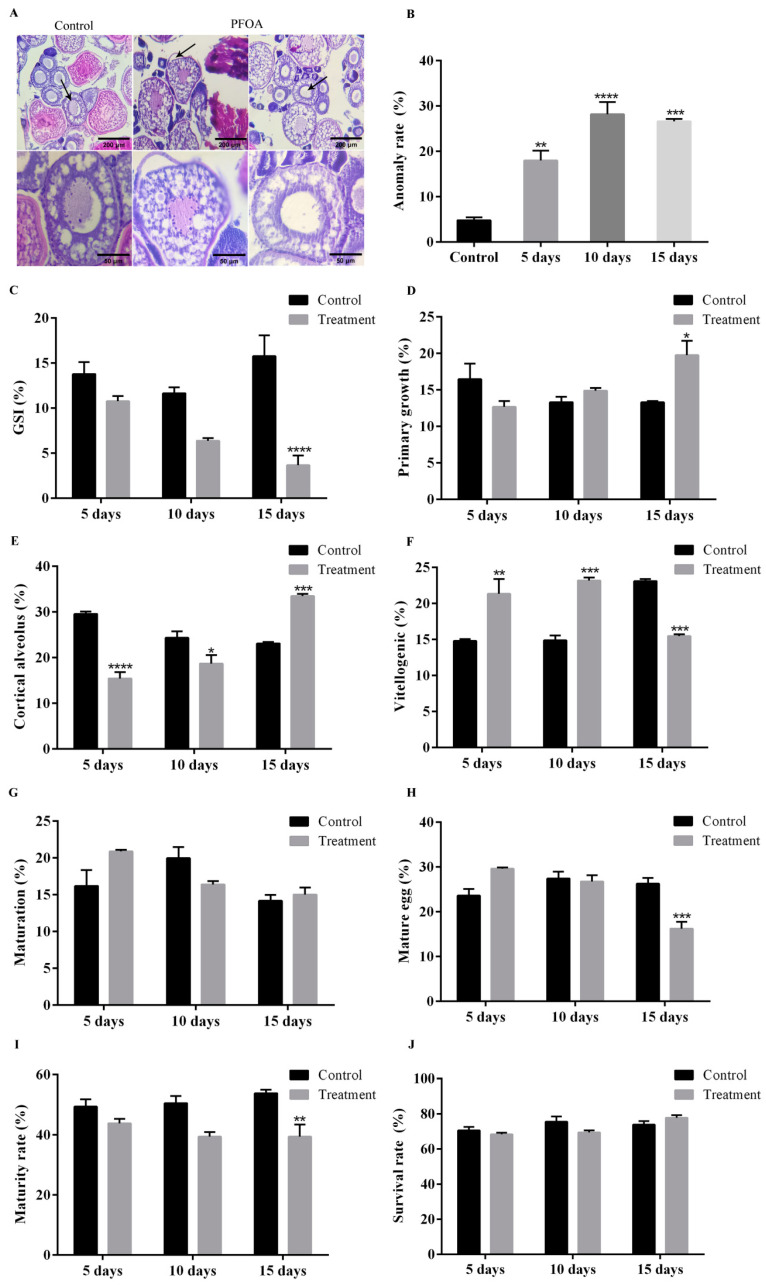
Effect of PFOA on the ovary morphology and oocytes maturation. (**A**) Representative images of histological changes in the ovaries of female zebrafish after exposure to PFOA, H&E-stained ovary section. From left to right, these are control, the loss of contact between the oocyte cell membranes and follicular cell layer, and vacuolation of the gonadosomatic tissue. Black arrows: enlarged area. (**B**) Percentage of abnormal oocytes in the ovary. (**C**) Effects of PFOA on gonadosomatic index (GSI). (**D**) Percentage of primary growth oocytes. (**E**) Percentage of cortical alveolus oocytes. (**F**) Percentage of vitellogenic oocytes. (**G**) Percentage of maturation oocytes. (**H**) Percentage of mature egg oocytes. (**I**) Effect of PFOA on oocytes maturation in in vitro culture. (**J**) Survival rates of zebrafish oocytes after in vitro culture. Data represent the mean ± SEM from at least three separate experiments (* *p* < 0.05, ** *p* < 0.01, *** *p* < 0.001, **** *p* < 0.0001); the vertical bar represents the standard error of mean (SEM).

**Figure 3 ijms-25-00136-f003:**
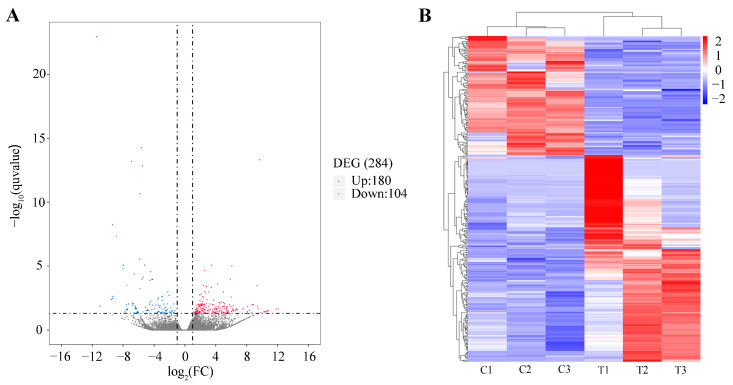
Overview of differently expressed genes between two groups. (**A**) Volcano plot showing the DEGs in control and PFOA-treated groups. The red dots reveal the up-regulated DEGs, the blue dots reveal the down-regulated DEGs, the grey dots represent non-differentially expressed genes. (**B**) Hierarchical cluster analysis of differentially expressed genes.

**Figure 4 ijms-25-00136-f004:**
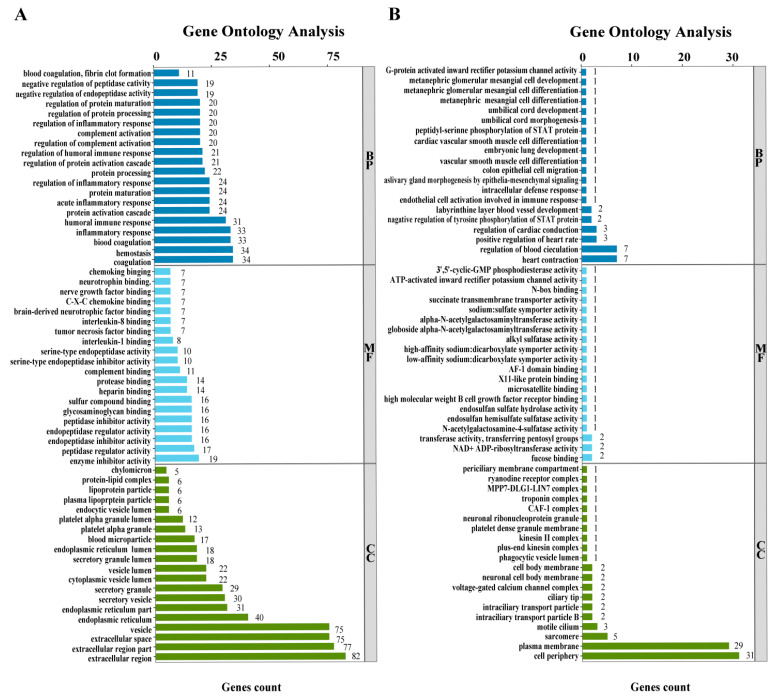
Functional enrichment analysis of DEGs. (**A**) Gene ontology analysis of the DEGs (up). (**B**) Gene ontology analysis of the DEGs (down).

**Figure 5 ijms-25-00136-f005:**
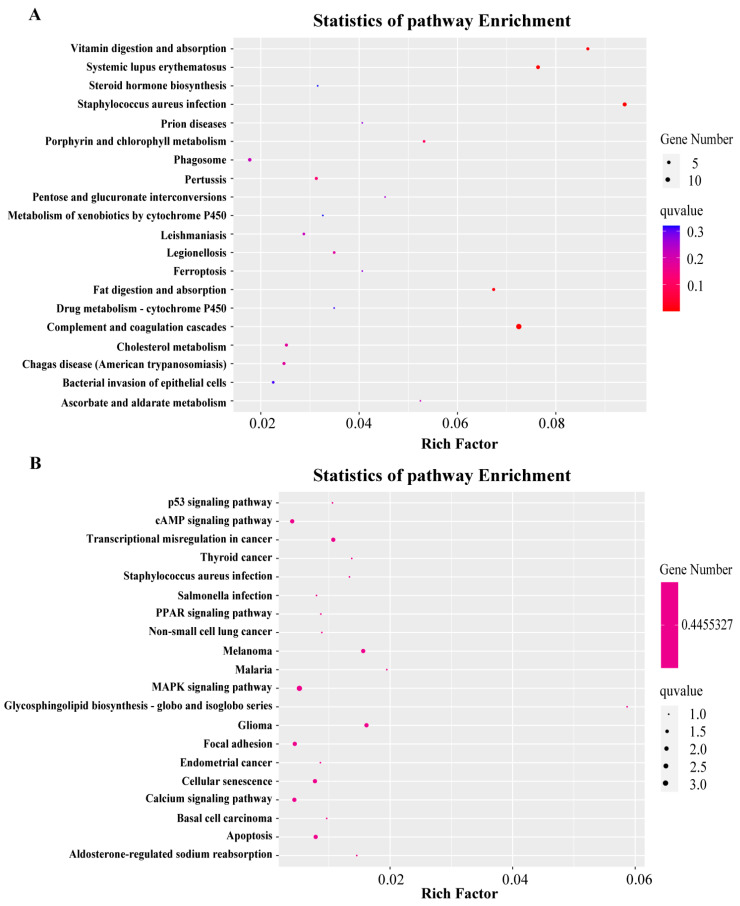
Statistics of Pathway Enrichment. (**A**) KEGG enrichment of the DEGs (up). (**B**) KEGG enrichment of the DEGs (down). The horizontal axis represents Rich factor, which refers to the proportion of enriched genes in a particular pathway compared to all genes present in that pathway. The vertical axis represents the enriched pathway, and the size of the dots represents the number of genes enriched in that pathway. Color represents enrichment significance, and the closer it is to red, the more significant it is.

**Figure 6 ijms-25-00136-f006:**
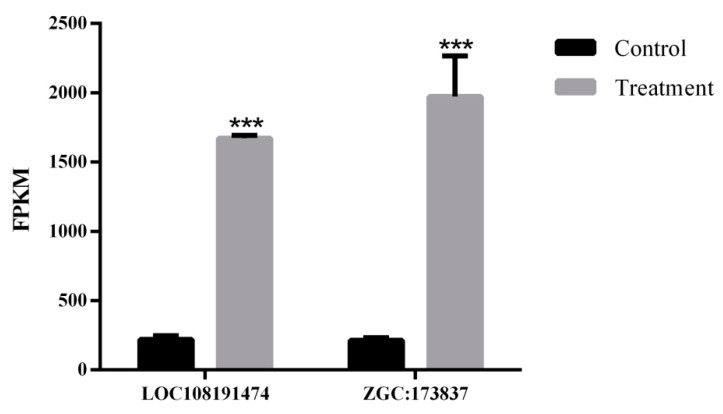
Expression levels of two immune-related genes in female zebrafish ovaries. Data represent mean ± SEM; the vertical bar represents the standard error of mean (SEM). Asterisks (***) indicate a statistically significant difference at *p* < 0.001.

**Table 1 ijms-25-00136-t001:** Acute toxicity test of PFOA on zebrafish.

Exposure Duration (h)	LC50 (mg/L)	95% Confidence Interval
24	423.654	394.104–450.290
48	410.980	385.357–441.295
72	393.741	359.000–418.490
96	375.578	347.802–404.769

**Table 2 ijms-25-00136-t002:** Effects of PFOA on fertility.

Concentration (mg/L)	Fertilization Rate (%)	Hatching Rate (%)	Deformation Rate (%)
0	78.00 ± 2.65	88.67 ± 2.19	0.33 ± 0.33
25	79.33 ± 1.20	90.33 ± 0.67	0.00 ± 0.00
50	76.33 ± 0.67	87.00 ± 4.04	2.33 ± 0.67
75	63.00 ± 6.00	81.00 ± 3.00	8.33 ± 0.88
100	51.67 ± 1.20	56.74 ± 1.43	13.61 ± 1.41

**Table 3 ijms-25-00136-t003:** Effect of PFOA on oocytes maturation.

	Day 5		Day 10		Day 15	
	Control	Treatment	Control	Treatment	Control	Treatment
GSI (%)	13.80 ± 1.33	10.77 ± 0.60	11.65 ± 0.68	6.40 ± 0.27	15.78 ± 2.32	3.68 ± 1.07
Primary growth (%)	16.47 ± 2.14	12.67 ± 0.80	13.33 ± 0.75	14.90 ± 0.36	13.30 ± 0.17	19.75 ± 1.97
Cortical alveolus (%)	29.59 ± 0.51	15.45 ± 1.40	24.35 ± 1.39	18.70 ± 1.86	23.12 ± 0.29	33.50 ± 0.50
Vitellogenic (%)	14.80 ± 0.26	21.35 ± 2.04	14.88 ± 0.67	23.21 ± 0.42	23.12 ± 0.29	15.48 ± 0.24
Maturation (%)	16.21 ± 2.16	20.88 ± 0.22	19.97 ± 1.51	16.40 ± 0.45	14.18 ± 0.82	15.02 ± 0.95
Mature egg (%)	23.61 ± 1.49	29.66 ± 0.26	27.46 ± 1.53	26.79 ± 1.35	26.27 ± 1.31	16.24 ± 1.55
Maturity rate (%)	49.44 ± 2.42	43.89 ± 1.47	50.56 ± 2.42	39.45 ± 1.47	53.89 ± 1.11	39.45 ± 4.00
Survival rate (%)	70.56 ± 2.00	68.33 ± 0.96	75.56 ± 2.94	69.44 ± 1.11	73.89 ± 2.00	77.78 ± 1.47

**Table 4 ijms-25-00136-t004:** Quality analysis of sequence after filtration.

Sample	Clean Reads	Clean Bases (bp)	GC (%)	Q20 (%)	Q30 (%)
Control.1	50,483,288	7,572,493,200	47.99; 48.06	96.79; 96.64	91.98; 91.46
Control.2	52,995,288	7,949,293,200	47.89; 47.89	96.95; 97.23	92.47; 92.88
Control.3	41,232,102	6,184,815,300	48.24; 48.33	96.92; 95.81	92.35; 89.66
Treatment.1	48,007,202	7,201,080,300	48.65; 48.77	96.77; 96.06	91.93; 90.26
Treatment.2	39,874,770	5,981,215,500	48.07; 48.13	96.84; 96.50	92.18; 91.23
Treatment.3	42,874,982	6,431,247,300	48.14; 48.16	96.95; 95.85	92.54; 89.81

## Data Availability

The datasets presented in this study can be found in online repositories. The names of the repository/repositories and accession number(s) can be found below: NCBI (accession: PRJNA932793).

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
