# Peer review of "Perfluorooctanoic Acid (PFOA) Exposure Compromises Fertility by Affecting Ovarian and Oocyte Development"

_ijms, 2023, doi:10.3390/ijms25010136_

Round 1
Reviewer 1 Report
Comments and Suggestions for Authors
Comments on this article.
This is a research article from Han Zhang et al.
This research paper deals with the effect of an emerging pollutant , the PFOA, on zebrafish's oocytes and their capability to be fertilized.
This is a relevant paper since PFOA is an emerging molecule and there is a lack of knowledge concerning its adverse effects in particular on reproduction and fertilization aspects.
In a traditional way, the manuscript is divided in introduction, Materials and Methods, Results and Discussion.
The introduction is short but clear and present the importance of working on PFOA. This part highlights the effects of PFOA, in particular the effects which concern aspects of reproduction and female gametes. The literature seems up to date. The authors could add data concerning PFOA in the environment (concentration, accumulation ...)
The Materials and Methods part presents the animal model (zebrafish) and the exposure procedure, and the many techniques used in this study, from morphological and histological aspects to transcriptomic ones.
The results part is the most developed. It is based of 6 figures and 2 tables. it's a very complete part. However, some aspect of transcriptomic study are not clear. Thus, figure 4 is of poor quality and therefore not readable. Is it possible de change this ? Or to present in another way the analysis (maybe in supplemented data ?) ? There is a contrast between the simplicity and clarity of Figures 1 to 3 and 6, compared to Figures 3, 4 and 5. Hopefully, the text and captions were clear enough to understand and showing the importance of - at least - two immune-related genes, when ovaries were exposed to PFOA.
Ethics statement : is there an approval number concerning animals housekeeping, exposure and use ?
Comments on the Quality of English LanguageNo comment on English language. Only minor editing of English are required.
The text is well written and clear.
Author Response
Thank you very much for taking the time to review this manuscript. I have carefully reviewed the manuscript and made revisions to address each of the issues you raised. I have marked all the changes in red. Your suggestions have been very helpful in improving the article, and I appreciate your valuable feedback. Please do not hesitate to let me know if you find any further deficiencies during the review process. I hope that the revised article will meet the publication standards of your journal soon. Thank you again for your time and effort in reviewing my work.
Comments 1: The introduction is short but clear and present the importance of working on PFOA. This part highlights the effects of PFOA, in particular the effects which concern aspects of reproduction and female gametes. The literature seems up to date. The authors could add data concerning PFOA in the environment (concentration, accumulation ...)
Response 1: Thank you for pointing this out. I agree with this comment. Therefore, I have added data on PFOA in lines 40-46 of the manuscript and highlighted it in red. The added content are "Among seven Chinese river basins, the maximum average concentration of PFOA was in the Yangtze River Basin (58 ng/L), while the lowest concentration was in the Songhua River Basin (< 1 ng/L) (Li, Wang et al. 2021). A study investigated the level of PFOA contamination in twenty-five samples of fresh fillet of five widely consumed fish species purchased from large retailers in Italy, it found PFOA was present in all samples, and concentration up to 487 ng/kg (mean = 75 ng/kg) (Barbarossa, Gazzotti et al. 2016)."
Comments 2: The results part is the most developed. It is based of 6 figures and 2 tables. it's a very complete part. However, some aspect of transcriptomic study are not clear. Thus, figure 4 is of poor quality and therefore not readable. Is it possible de change this ? Or to present in another way the analysis (maybe in supplemented data ?) ? There is a contrast between the simplicity and clarity of Figures 1 to 3 and 6, compared to Figures 3, 4 and 5. Hopefully, the text and captions were clear enough to understand and showing the importance of - at least - two immune-related genes, when ovaries were exposed to PFOA.
Response 2: Thank you for providing me with your valuable feedback on my figure. Your serious and rigorous approach is truly appreciated and I have learned a lot from it. I have made the necessary modifications to Figure 4 as per your suggestions and have now inserted it in line 182 of my manuscript.
Comments 3: Ethics statement : is there an approval number concerning animals housekeeping, exposure and use ?
Response 3: Yes. All experimental and sampling procedures were conducted according to the Guidelines of the Animal Care of the Zhejiang Ocean University and were approved by the Ethics Committee of the university. I have added it to lines 365 - 367 of the manuscript and highlighted it in red.
Reviewer 2 Report
Comments and Suggestions for Authors
The paper titled “Perfluorooctanoic acid (PFOA) exposure compromises fertility via affecting ovarian and oocyte development” reports several data related to effect induced by PFOA, using an important model system, zebrafish. Specifically, reproductive toxicity was studied. In particular, authors report data about the outcome of fertilization and hatching. The authors also investigated ovarian damage and oocyte maturation.
The research is adequately conducted, the materials and methods are complete and the results are well illustrated and clearly discussed.
I suggest minor revisions.
Figure 1.
The authors should enlarge the photos acquired under the microscope (about double), as it is difficult to visualize the details. The value of the bar is also unreadable.
In the Y axes, in the Y axis add a space between rate and (%)
In D, a complete scale of % should be reported.
Figure 2.
C) add a space before (%), from C to I. Also in this case a different % scale is reported. Please, standardize.
Figure 3
A and B. the size of characters should be increased.
The conclusions should be a little broader. The authors could report what the results of this research add to the existing literature.
Author Response
Thank you very much for taking the time to review this manuscript. I have carefully reviewed the manuscript and made revisions to address each of the issues you raised. I have marked all the changes in red. Your suggestions have been very helpful in improving the article, and I appreciate your valuable feedback. Please do not hesitate to let me know if you find any further deficiencies during the review process. I hope that the revised article will meet the publication standards of your journal soon. Thank you again for your time and effort in reviewing my work.
Comments 1: Figure 1.The authors should enlarge the photos acquired under the microscope (about double), as it is difficult to visualize the details. The value of the bar is also unreadable.In the Y axes, in the Y axis add a space between rate and (%)In D, a complete scale of % should be reported.
Response 1: Thank you for pointing this out. I agree with this comment. Therefore, I inserted the modified image in line 106 of the article resolving the issue you raised in Figure 1. Table 2 has been added in line 105 of the manuscript to provide a complete scale of % presented in Figure 1.
Comments 2: Figure 2.C) add a space before (%), from C to I. Also in this case a different % scale is reported. Please, standardize.
Response 2: After carefully reviewing your feedback, I have made modifications to Figure 2 and inserted it into line 140 of the manuscript, as per your suggestions. In addition, I added Table 3 on line 139 of the manuscript to explain the data in Figure 2.
Comments 3: Figure 3 A and B. the size of characters should be increased.
Response 3: Thank you for providing me with your valuable feedback on my figure. Your serious and rigorous approach is truly appreciated and I have learned a lot from it. I have made the necessary modifications to Figure 3 as per your suggestions and have now inserted it in line 169 of my manuscript. Additionally, I have also made the required font changes to ensure consistency throughout the document.
Comments 4: The conclusions should be a little broader. The authors could report what the results of this research add to the existing literature.
Response 4: Thank you for pointing out the shortcomings in my manuscript. I agree with this comment. Therefore, I have added some content to the conclusion. For your convenience in reading, I have highlighted the new content in red in the conclusion, which can be found on line 369-371 of the manuscript. The added content are “Previous studies have shown that PFOA has reproductive toxicity. However, the potential mechanism of PFOA exposure on fertility remains unclear. We found exposure to PFOA can cause an inflammatory response that can lead to ovarian damage and delayed oocyte development. This study's findings provide valuable insights that can contribute to future research on the potential mechanisms of PFOA exposure on fertility.”
Reviewer 3 Report
Comments and Suggestions for Authors
The manuscript describes the perfluorooctanoic acid (PFOA) exposure compromises fertility via affecting ovarian and oocyte developmen. The topic is relevant to the aim and scope of the International Journal of Molecular Sciences. The manuscript is well written and easy to follow. Some clarifications in the texts are needed. Overall, this manuscript meets the standard for acceptance after addressing the below comments:
1) According to Table 1, LC50 at 96 h was 375.578 mg/l. And in the second paragraph of 2.2 Exposure regimens, the highest exposure group concentration was 1/4 of the 96h LC50. Then, the concentration was corresponding to 90 mg/l, not 100 mg/l?
2) Why should BSA concentration be 0.5 mg/ml in in vitro culture? And what was the role of BSA, although L-15 medium provided mimetic environment to ovary? Why was % of PS lowered in maturation medium?
3) Why was deformation rate higher than the decrease in hatching rate?
4) In Figure 2, why was vitellogenic the lowest at 15 days than other days? And why was the vitellogenic at 15 days only lower than control?
5) Figure 3 and 4 are not readable. Please modify the figures.
Author Response
Thank you very much for taking the time to review this manuscript. I have carefully reviewed the manuscript and made revisions to address each of the issues you raised. I have marked all the changes in red. Your suggestions have been very helpful in improving the article, and I appreciate your valuable feedback. Please do not hesitate to let me know if you find any further deficiencies during the review process. I hope that the revised article will meet the publication standards of your journal soon. Thank you again for your time and effort in reviewing my work.
Comments 1: According to Table 1, LC50 at 96 h was 375.578 mg/l. And in the second paragraph of 2.2 Exposure regimens, the highest exposure group concentration was 1/4 of the 96h LC50. Then, the concentration was corresponding to 90 mg/l, not 100 mg/l?
Response 1: Thank you for providing your valuable feedback on my manuscript. When I was designing the experiment, I think that the highest concentration of exposure for the group would be around 1/4 of the 96-hour LC50 value, rather than an exact value. This was my mistake, which led to a lack of precision in the experiment. I appreciate you bringing this to my attention. During the experiment, no fish died at a concentration of 100mg/L, leading me to overlook the issue of concentration setting. Thank you for your careful reading and guidance.
Comments 2: Why should BSA concentration be 0.5 mg/ml in in vitro culture? And what was the role of BSA, although L-15 medium provided mimetic environment to ovary? Why was % of PS lowered in maturation medium?
Response 2: Thank you for pointing this out. 1: In animal cell culture without serum, BSA can provide mechanical, physiological and nutritional benefits We have referred to the article of Xu Peng fei to determine the appropriate concentration in 0.5 mg/ml.
2:It has been observed that PS plays a crucial role in inhibiting bacteria in cell culture. After consulting a large number of literature, it has been concluded that the appropriate concentration of PS in cell culture is 1%. As a result, the PS concentration in mature culture medium has been set to 1%. However, during our experiment, it was found that if the PS concentration in OCM is also 1%, the cells are vulnerable to bacterial infection during subsequent culture. This susceptibility could be due to bacterial infection received during cell selection. We consulted the reagent company and conducted a control experiment, ultimately determining that the concentration of PS in OMM was 3%
Comments 3: Why was deformation rate higher than the decrease in hatching rate?
Response 3: Thank you for pointing this out. Previous studies have reported negative impacts of PFOA on the reproductive capacity of zebrafish, including reduced hatching rates and egg production. We observed deformities in zebrafish during hatching, which has not been previously reported. But in the exposure to BPA, we found similar results. In report on BPA, that high concentrations (50μm) of BPA can result in a mortality rate increase of 50% to 60%. When the BPA concentration is at or above 30μm, the likelihood of deformities also increases. At 50μm, the deformity rate exceeds 90%. We think that PFOA may cause a decrease in hatching rate and an increase in deformation rate, but there is no direct correlation between these two values. (Bisphenol A Exposure Induces Sensory Processing Deficits in Larval Zebrafish during Neurodevelopment).
Comments 4: In Figure 2, why was vitellogenic the lowest at 15 days than other days? And why was the vitellogenic at 15 days only lower than control?
Response 4: Thank you for carefully inspecting and providing valuable feedback.The ovary of zebrafish is in fact asynchronous and oocytes at different stages of development are simultaneously present. Our research indicates that exposure to PFOA can impede the growth and development of oocytes. This impeded growth causes a majority of the oocytes to remain in primary growth and cortical alveolar phase, which results in a reduced number of mature oocytes and hampers reproductive ability.
Comments 5: Figure 3 and 4 are not readable. Please modify the figures
Response 5: Thank you for your valuable suggestions regarding the manuscript. I have incorporated your feedback and made modifications to Figure 3 and Figure 4. You can find the updated Figure 3 on line 169 of the manuscript and the updated Figure 4 on line 182.
Round 2
Reviewer 3 Report
Comments and Suggestions for Authors
All of the issues have been addressed.